# Efficient Bayesian Optimization with Deep Kernel Learning and Transformer Pre-trained on Muliple Heterogeneous Datasets

## Abstract

Bayesian optimization (BO) is widely adopted in black-box optimization problems and it relies on a surrogate model to approximate the black-box response function. With the increasing number of black-box optimization tasks solved and even more to solve, the ability to learn from multiple prior tasks to jointly pre-train a surrogate model is long-awaited to further boost optimization efficiency. In this paper, we propose a simple approach to pre-train a surrogate, which is a Gaussian process (GP) with a kernel defined on deep features learned from a Transformer-based encoder, using datasets from prior tasks with possibly heterogeneous input spaces. In addition, we provide a simple yet effective mix-up initialization strategy for input tokens corresponding to unseen input variables and therefore accelerate new tasks' convergence. Experiments on both synthetic and real benchmark problems demonstrate the effectiveness of our proposed pre-training and transfer BO strategy over existing methods.

## 1 Introduction

In black-box optimization problems, one could only observe outputs of the function being optimized based on some given inputs, and can hardly access the explicit form of the function. These kinds of optimization problems are ubiquitous in practice (e.g., (Mahapatra et al., 2015; Korovina et al., 2020; Griffiths & Lobato, 2020)). Among black-box optimization problems, some are particularly challenging since their function evaluations are expensive, in the sense that the evaluation either takes a substantial amount of time or requires a considerable monetary cost. To this end, Bayesian Optimization (BO; Shahriari et al. (2016)) was proposed as a sample-efficient and derivative-free solution for finding an optimal input value of black-box functions.

BO algorithms are typically equipped with two core components: a surrogate and an acquisition function. The surrogate is to model the objective function from historical interactions, and the acquisition function measures the utility of gathering new input points by trading off exploration and exploitation. Traditional BO algorithms adopt Gaussian process (GP; Rasmussen & Williams (2009)) as the surrogates, and different tasks are usually optimized respectively in a cold-start manner. In recent years, as model pre-training showed significant improvements in both convergence speed and prediction accuracy (Szegedy et al., 2016; Devlin et al., 2019), pre-training surrogate(s) in BO becomes a promising research direction to boost its optimization efficiency.

Most existing work on surrogate pre-training (Bardenet et al., 2013; Swersky et al., 2013; Yogatama & Mann, 2014; Springenberg et al., 2016; Wistuba et al., 2017; Perrone et al., 2018; Feurer et al., 2018a; Wistuba & Grabocka, 2021) assumes that the target task shares the same input search space with prior tasks generating historical datasets. If this assumption is violated, the pre-trained surrogate cannot be directly applied and one has to conduct a cold-start BO. Such an assumption largely restricts the scope of application of a pre-trained surrogate, and also prevents it from learning useful information by training on a large number of similar datasets. To overcome these limitations, a text-based method was proposed recently. It formulates the optimization task as a sequence modeling problem and pre-trains a single surrogate using various optimization trajectories (Chen et al., 2022).

In this work, we focus on surrogate pre-training that transfers knowledge from prior tasks to new ones with possibly different input search spaces, for further improving the optimization efficiency

of BO. We adopt a combination of Transformer (Vaswani et al., 2017) and deep kernel Gaussian process (Wilson et al., 2016b) for the surrogate, which enables joint training on prior datasets with variable input dimensions. For a target task, only the feature tokenizer of the pre-trained model needs to be modularized and reconstructed according to its input space. Other modules of the pre-trained model remain unchanged when applied to new tasks, which allows the new task to make the most of prior knowledge. Our contributions can be summarized as follows:

- To the best of our knowledge, this is the first transfer BO method that is able to jointly pre-train on tabular data from tasks with heterogeneous input spaces.

- We provide a simple yet effective strategy of transferring the pre-trained model to new tasks with previously unseen input variables to improve optimization efficiency.

- Our transfer BO method shows clear advantage on both synthetic and real problems from different domains, and also achieves the new state-of-the-art results on the HPO-B (Pineda-Arango et al., 2021) public datasets.

## 2 BACKGROUND

**Gaussian process**   A Gaussian process is a collection of random variables, any finite number of which have a joint Gaussian distribution (Rasmussen & Williams, 2009). Formally, a GP is represented as $f(\boldsymbol{x}) \sim \mathcal{GP}(m(\boldsymbol{x}), k(\boldsymbol{x}, \boldsymbol{x}'))$, where $m(\boldsymbol{x})$ and $k(\boldsymbol{x}, \boldsymbol{x}')$ denotes mean and covariance function, respectively. Given a dataset $\mathcal{D} = \{(\boldsymbol{x}^{(i)}, y^{(i)})\}_{i=1}^n$ with $n$ examples, any collection of function values has a joint Gaussian distribution $\boldsymbol{f} = [f(\boldsymbol{x}^{(1)}), \ldots, f(\boldsymbol{x}^{(n)})]^\top \sim \mathcal{N}(\boldsymbol{\mu}, \boldsymbol{K}_{\mathbf{x},\mathbf{x}})$, where the mean vector $\boldsymbol{\mu}_i = m(\boldsymbol{x}^{(i)})$, $[\boldsymbol{K}_{\mathbf{x},\mathbf{x}}]_{ij} = k(\boldsymbol{x}^{(i)}, \boldsymbol{x}^{(j)})$. A nice property of GP is that its distributions of various derived quantities can be obtained explicitly. Specifically, under the additive Gaussian noise assumption, the predictive distribution of the GP evaluated at a new test example $\boldsymbol{x}^{(*)}$ can be derived as

$$p(\boldsymbol{f}^{(*)} | \boldsymbol{x}^{(*)}, \mathcal{D}) \sim \mathcal{N}(\mathbb{E}[\boldsymbol{f}^{(*)}], \text{cov}(\boldsymbol{f}^{(*)})), \tag{1}$$

where $\mathbb{E}[\boldsymbol{f}^{(*)}] = m(\boldsymbol{x}^{(*)}) + \boldsymbol{K}_{\boldsymbol{x}^{(*)},\mathbf{x}}[\boldsymbol{K}_{\mathbf{x},\mathbf{x}} + \sigma^2 \boldsymbol{I}]^{-1}\boldsymbol{y}$, $\text{cov}(\boldsymbol{f}^{(*)}) = k(\boldsymbol{x}^{(*)}, \boldsymbol{x}^{(*)}) - \boldsymbol{K}_{\boldsymbol{x}^{(*)},\mathbf{x}}[\boldsymbol{K}_{\mathbf{x},\mathbf{x}} + \sigma^2 \boldsymbol{I}]^{-1}\boldsymbol{K}_{\mathbf{x},\boldsymbol{x}^{(*)}}$, $\boldsymbol{K}_{\boldsymbol{x}^{(*)},\mathbf{x}}$ denotes the vector of covariances between the test example $\boldsymbol{x}^{(i)}$ and the $n$ training examples, and $\boldsymbol{y}$ is the vector consisting of all response values.

**Bayesian Optimization**   Bayesian optimization (Shahriari et al., 2016) uses a probabilistic surrogate model for data-efficient black-box optimization. It is suited for *expensive black-box optimization*, where objective evaluation can be time-consuming or of high cost. Given the previously gathered dataset $\mathcal{D}$, BO uses surrogate models like GP to fit the dataset. For a new input $\boldsymbol{x}^{(*)}$, the surrogate model gives predictive distribution in equation 1, then an acquisition function is constructed with both the prediction and uncertainty information to balance exploitation and exploration. The acquisition is optimized by third-party optimizer like evolutionary algorithm to generate BO recommendation. Throughout this paper, we use the lower confidence bound (LCB) Srinivas et al. (2009) as the acquisition function. $\text{LCB}(\boldsymbol{x}) = m(\boldsymbol{x}) - \kappa \times \sigma(\boldsymbol{x})$, where $\sigma(\boldsymbol{x})$ denotes the standard deviation and $\kappa$ (set to 3 in experiments) is a constant for tuning the exploitation and exploration trade-off.

**FT-Transformer**   FT-Transformer (Gorishniy et al., 2021) is a recently proposed attention-based model for tabular data modeling. It consists of a Feature-Tokenizer layer, multiple Transformer layers, and a prediction layer. The Feature-Tokenizer layer enables its ability of handling tabular data. For $d$ numerical input features $\boldsymbol{x} = [x_1, \ldots, x_d]^\top$, the Feature-Tokenizer layer initializes a value-dependent embedding table $\boldsymbol{W} \in \mathbb{R}^{d \times d_e}$ and a column-dependent embedding table $\boldsymbol{B} \in \mathbb{R}^{d \times d_e}$, where $d_e$ is the dimension of embedding vector. During forward-pass of the Feature-Tokenizer layer, the $i$-th feature $x_i$ would be transformed to $x_i \times \boldsymbol{w}_i + \boldsymbol{b}_i$, where $\boldsymbol{w}_i$ and $\boldsymbol{b}_i$ are the $i$-th row in $\boldsymbol{W}$ and $\boldsymbol{B}$. In this way, an $n \times d$ matrix is transformed into a $n \times d \times d_e$ tensor. Then, a `[CLS]` token embedding is appended to the tensor and the tensor is passed to the stacked transformer layers to extract output embedding vectors. The output embedding vector corresponding to the `[CLS]` token is used as the output representation. The output representation is then passed into the prediction layer for final model prediction. The tokenization process for categorical data is implemented by a look-up table, in which each categorical variable corresponds to a $\boldsymbol{b}_i$ and each unique value of a variable corresponds to a $\boldsymbol{w}_i$.

We use FT-Transformer as the backbone of our method. Throughout this paper, we only consider numerical features, however as FT-Transformer can also handle categorical features, our method can be easily extended to mixed search space with both numerical and categorical features.

## 3 METHODOLOGY

### 3.1 PROBLEM SETTING

Given a target function $f_T(\boldsymbol{x}) : \mathbb{R}^{d_T} \to \mathbb{R}$ where $d_T$ is the dimension of $f_T$, we would like to apply Bayesian optimization to find its minimizer:

$$\boldsymbol{x}_* = \mathrm{argmin}_{\boldsymbol{x}} f_T(x)$$

Assume we have $N$ source dataset $\{\mathcal{D}_1^S, \ldots, \mathcal{D}_N^S\}$ where $\mathcal{D}_i^S = \{X_i^S, \boldsymbol{y}_i^S\}$, $X_i^S \in \mathbb{R}^{N_i \times d_i}$ and $y_i^S \in \mathbb{R}^{N_i}$, we want to pre-train a surrogate model on all these $N$ historical datasets to accelerate the BO convergence on target task.

If $d_1 = d_2 = \ldots d_N = d_T$, and the feature names are aligned, pre-training methods like (Wistuba & Grabocka, 2021; Wistuba et al., 2016; Feurer et al., 2018b) can be applied. However, if either the dimension or parameter names are unaligned, pre-training and fine-tuning become non-trivial.

Taking hyper-parameter optimization (HPO) for AutoML as an example. Suppose we have the following two historical HPO records:

- HPO for Random forest on dataset A, where `max_features`, `max_depth` are tuned,
- HPO for LightGBM on dataset B, where `learning_rate` and `reg_alpha` are tuned,

now we want to perform HPO for XGBoost model on a new dataset C, the hyper-parameters to be tuned are `learning_rate`, `max_depth` and `col_sample_by_level`, and we want to use the two source datasets to pre-train the surrogate model for BO.

### 3.2 MULTI-SOURCE DEEP KERNEL LEARNING WITH FT-TRANSFORMER

The basic idea of our method is to use FT-Transformer (Gorishniy et al., 2021) as the feature extractor of deep kernel Gaussian processes (FT-DKL) and pre-train the FT-DKL on similar source tasks. Given that transformer can handle variable-length input, we can use the FT-DKL to jointly pre-train on multiple heterogeneous datasets with unaligned parameter spaces, making the FT-Transformer a *multi-source* FT-Transformer.

The first step of our algorithm is data normalization. When there are multiple unaligned source datasets, we independently normalize the objective of each source dataset. As for the normalization of input features, if two source datasets share common features, the shared common features are merged and jointly normalized.

After the source datasets are normalized, we initialize a multi-source FT-Transformer, where the embedding table in the Feature-tokenizer layer corresponds to the union of features of all source datasets. Following the common procedures of training DKL models, we firstly pre-train the multi-source FT-Transformer with linear output layer and MSE loss, and then replace the linear output layer with a sparse variational Gaussian process (SVGP) layer and pre-train the model with ELBO loss, in this stage, the weight of FT-Transformers, the GP hyper-parameters and variational parameters are jointly updated. Details about SVGP and sparse varational deep kernel learning can be seen in (Hensman et al., 2015; Wilson et al., 2016a).

After pre-training, we can transfer the model to downstream target optimization task, where the transformer layers, sparse Gaussian process layer and the `[CLS]` embedding of Feature-Tokenizer layer are copied to initialize the target FT-DKL model.

For the target task parameters already seen in source tasks, the corresponding source embedding vectors are also copied to the target FT-DKL model; for the unseen target task parameters, we use mix-up initialization: we randomly select two embedding vectors $\boldsymbol{e}_1$ and $\boldsymbol{e}_2$ from source embeddings and a random number $\alpha \sim \mathcal{U}(0, 1)$, the new embedding vector is initialized as

$$\boldsymbol{e} = \alpha \times \boldsymbol{e}_1 + (1 - \alpha) \times \boldsymbol{e}_2.$$

After the target FT-DKL is initialized, we directly fine-tune the target FT-DKL model with target data using ELBO loss at the start of each BO iteration. The fine-tuned FT-DKL model can be used as a regular Gaussian process to construct acquisition functions and give recommendations. We summarize our algorithm in Algorithm 1. In Section A.1, we provide an illustrative example to demonstrate the proposed model architecture.

---

**Algorithm 1** Pre-training and fine-tuning of multi-source FT-DKL

---

**Require:** source datasets $\{D_S^1, \ldots, D_S^N\}$
    Jointly normalize the source datasets
    Construct multi-source FT-Transformer with the union of all source features for the Feature-Tokenizer layer
    Joint training of multi-source datasets with MSE loss
    Joint training of multi-source with GP layer and ELBO loss
    **while** target task not finished **do**
        Transfer the source FT-DKL to target FT-DKL with embedding mix-up
        Fine-tune the target FT-KL with ELBO loss
        Use the predictive distribution of FT-DKL to construct acquisition function
        Optimize the acquision function for BO recommendation
    **end while**

---

## 4   Related Work

Warm-starting a new BO task by knowledge transfer from prior tasks can significantly boost optimization efficiency. One of the most common transfer approaches is to pre-train a surrogate model on the entire prior data. Earlier methods considered learning a joint GP-based surrogate model on combined prior data (Bardenet et al., 2013; Swersky et al., 2013; Yogatama & Mann, 2014), which may easily suffer from high computational complexity when applied to large data sets. As a result, some later works tried to alleviate this issue by using multi-task learning (Perrone et al., 2018) or ensembles of GP, where a GP was learned for each task (Wistuba et al., 2017; Feurer et al., 2018a). Neural network models have also been adopted to either learn a task embedding for each task inside a Bayesian neural network (Springenberg et al., 2016), or pre-train a deep kernel model and fine-tune on target tasks (Wistuba & Grabocka, 2021).

Besides the surrogate model, learning new acquisition functions is another valid approach for knowledge transfer (Volpp et al., 2020). However, all of these existing works considered a fixed search space. In other words, a new task with a different search space would fail in directly borrowing strength from prior data sets. Most recently, a text-based pre-trained model called OptFormer (Chen et al., 2022) was introduced to address this issue by adopting a Transformer model as the surrogate. Although both OptFormer and our work are able to pre-train using heterogeneous datasets, our work directly conducts learning on tabular data whereas OptFormer requires an additional tokenization step for the optimization trajectories.

Transformer (Vaswani et al., 2017) was initially proposed for machine translation. It was later adopted, beyond natural language processing, in computer vision (Dosovitskiy et al., 2021; Parmar et al., 2018), reinforcement learning (Chen et al., 2021; Zheng et al., 2022), etc. Because of its success in various fields, Transformer was also restructured for tabular data (Huang et al., 2020; Gorishniy et al., 2021) as another strong (neural network) baseline besides multi-layer perceptron.

## 5   Experiments

In this section, we demonstrate the high sample efficiency of our pre-trained FT-DKL with three experiments, including a high-dimensional synthetic function, HPO on the HPO-B (Pineda-Arango et al., 2021) benchmark, and a real-world wireless network optimization (WNO) problem. For the synthetic function and wireless network optimization problem, we transferred knowledge from a single low-dimensional source problem to a high-dimensional target task, while for the HPO-B benchmarks, 758 source tasks with dimensions ranging from 2 to 18 were merged and pre-trained,

the pre-trained model was fine-tuned for 86 different target HPO problems. The statistics of these experiments can be seen in Table 1.

Table 1: Statistics of experimental datasets

| Task | Source tasks | Source dimension | Source evaluations | Target tasks | Target dimension |
|---|---|---|---|---|---|
| Synthetic | 1 | 20 | 800 | 1 | 30 |
| HPO-B | 758 | 2-18 | 3,279,050 | 86 | 2-18 |
| WNO | 1 | 48 | 1800 | 1 | 69 |

All experiments were conducted on a server with eight-core `Intel(R) Xeon(R) Gold 6134` CPU and one `Quadro RTX 5000` GPU.

## 5.1 SCALED AND SHIFTED HIGH DIMENSIONAL ACKLEY FUNCTION OPTIMIZATION

In this section, we use the Ackley function with input scaling and offset as a demonstration. Ackley function is a popular benchmark function for black-box optimization algorithms, To make the function harder to optimize and transfer, we introduced random offset and scaling to each dimension. To do that, we firstly scaled the original search space to $[-1, 1]^D$ where $D$ was the input dimension; then we modified the original Ackley function as shown in Eq 2. In our experiment, we set $D = 30$ for the target optimization task.

$$
\begin{aligned}
f_D(\boldsymbol{x}) &= \text{Ackley}(s_1 \times (x_1 - o_1) \dots s_D \times (x_D - o_D)) \\
s_i &\sim \mathcal{U}(0.01, 2), i = 1, 2, \dots, D \\
o_i &\sim \mathcal{U}(-0.8, 0.8), i = 1, 2, \dots, D
\end{aligned}
\tag{2}
$$

Firstly, we compared GP-UCB on the original Ackley function and the modified Ackley function. As shown in Figure 1a, we see that although GP-UCB was able to reach a near-optimal solution for the original Ackley function, it failed to optimize the Ackley function with input scaling and offset.

We then compared different surrogate models on the modified Ackley function. For our pre-trained FT-DKL, we used the same modified Ackley function but with $D = 20$ as the source task, leaving 10 target task parameters unseen in the source task. We ran evolutionary algorithm on the 20-dimensional source task, and sampled 800 data points from the optimization trajectory to pre-train the FT-DKL model.

The following models were compared. For all surrogate models, lower confidence bound (LCB) was used as the acquisition function. We randomly sampled 5 points to initialize BO, except for GP-50, where 50 initial random points were used. For each model, BO was repeated five times to average out random fluctuations.

- GP, where Gaussian process with Matern3/2 kernel was used for BO,

- FT-DKL, where the FT-DKL model was used as the surrogate model, *without* any pre-training and model transfer,

- GP-50, Gaussian process as surrogate model, but with 50 points as random initialization,

- Pretrained-FT-DKL, FT-DKL pre-trained on 20-D data used as surrogate model.

As shown in Figure 1b, we can see that with only five points as random initialization, both GP and FT-DKL failed to optimize the AckleyOffsetScale function. With 50 points as random initialization, GP was able to find some low-regret solutions, but the result was far from optimal. On the other hand, when we were able to pre-train the FT-DKL on the 20-D source function and fine-tune the model on the 30-D target function, the pre-trained FT-DKL outperformed all other models significantly.

In this paper, we only consider how model performance is improved by multi-source pre-training. However, from a practical point of view, a more effective approach for transferring from a 20-D function to a 30-D function would be to directly copy the optimal values of the 20 dimensions, and only optimize the rest 10 dimensions. In Appendix A.3, we show that our proposed approach still outperformed Gaussian processes and FT-DKL when fixing the first 20 dimensions.

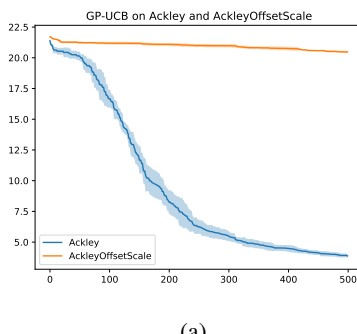 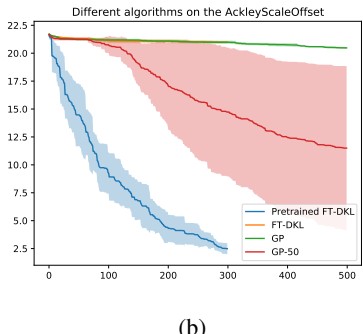

(a)                                             (b)

Figure 1: (a) GP-UCB on Ackley and the Ackley with scale and offset, it can bee seen that the Ackley with input scaling and offset is much harder to optimize; (b) Comparison of different algorithms on the Ackley with scale and offset.

## 5.2 HYPER-PARAMETER OPTIMIZATION USING HPO-B BENCHMARK

In this section, we demonstrate our method on HPO problems with the HPO-B benchmark (Pineda-Arango et al., 2021). HPO-B is the largest public benchmark for HPO, containing more than 1900 HPO-B tasks, with 176 different search spaces. We used the `HPO-B-v3` version in our experiment, where tasks related to the most frequent 16 search spaces were extracted and splitted into `meta-train-dataset`, `meta-validation-dataset`, and `meta-test-dataset`. After the train-validation-test splitting, there were 758 HPO tasks with 3,279,050 hyper-parameter configuration evaluations with 16 search spaces in the `meta-train-dataset`, and 86 tasks in the `meta-test-dataset`.

We jointly pre-trained the FT-DKL model on all the 758 meta-train tasks and then transferred the model to the target task during the optimization of meta-test tasks. We didn't use the meta validation dataset in this experiment. We pre-trained the model for 300 epochs with MSE loss and 50 epochs with ELBO loss. Unlike FSBO (Wistuba & Grabocka, 2021) where a distinct model was pre-trained for each search space, we used one common model with shared transformer layers for all the 758 source tasks. During pre-training, the meta-features like `space-ID` and `dataset-ID` were treated as categorical features to augment the dataset.

We followed the evaluation protocol of HPO-B, where five different sets of initialization points were provided by the benchmark for repeated experiments. We compared our algorithm against the results provided by the HPO-B benchmark, the following transfer and non-transfer BO methods were compared:

- Random: Random search
- GP: Gaussian processes
- DNGO (Snoek et al., 2015): Bayesian linear regression with feature extracted by neural networks,
- DGP (Wilson et al., 2016b): Gaussian process with deep kernel,
- BOHAMIANN (Springenberg et al., 2016): BO with BNN surrogate, trained with adaptive SGDHMC
- TST (Wistuba et al., 2016), RGPE (Feurer et al., 2018b), TAF (Wistuba et al., 2017): Transfer learning by weighted combination of Gaussian processes trained on source tasks with same design spaces,
- FSBO (Wistuba & Grabocka, 2021): Few-shot Bayesian optimization where deep kernel GP was pre-trained on source tasks that share the same design space with target task.

The result of normalized regret and average rank is shown in Figure 2 and Table 2. As can be seen, our pre-trained model outperformed all other reported results by a large margin, both with regard to average rank and average regret, throughout all the iterations, the immediate regret remained 34%-65% of the second best algorithm FSBO. Among all algorithms, our method was the only one that

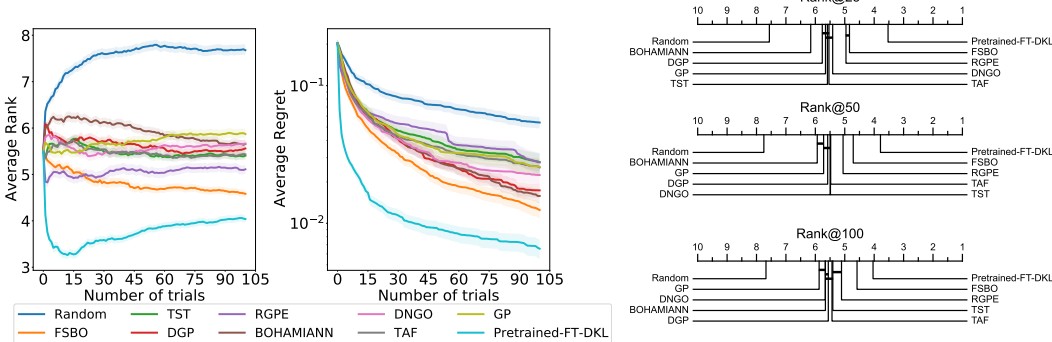

Figure 2: Comparisons of normalized regret and average ranks across all search spaces.

Table 2: Normalized regret comparison at different iterations

| Algorithms | Regret@1 | Regret@5 | Regret@15 | Regret@30 | Regret@50 | Regret@100 |
|---|---|---|---|---|---|---|
| Random | 0.189 | 0.142 | 0.102 | 0.082 | 0.072 | 0.0540 |
| GP | 0.181 | 0.106 | 0.064 | 0.044 | 0.035 | 0.0258 |
| TST | 0.164 | 0.104 | 0.064 | 0.047 | 0.039 | 0.0278 |
| DNGO | 0.165 | 0.099 | 0.057 | 0.041 | 0.028 | 0.0224 |
| TAF | 0.164 | 0.101 | 0.060 | 0.043 | 0.036 | 0.0254 |
| DGP | 0.183 | 0.099 | 0.061 | 0.041 | 0.028 | 0.0173 |
| BOHAMIANN | 0.173 | 0.118 | 0.061 | 0.041 | 0.028 | 0.0158 |
| RGPE | 0.140 | 0.104 | 0.064 | 0.053 | 0.046 | 0.0278 |
| FSBO | 0.152 | 0.087 | 0.049 | 0.032 | 0.021 | 0.0105 |
| Pretrained-FT-DKL | **0.066** | **0.030** | **0.017** | **0.011** | **0.009** | **0.0065** |

achieved a regret of less than 0.1 after only one iteration and less than 0.01 after 100 iterations. The per-space comparison of normalized regret and average rank is put in Appendix A.4, where pre-trained FT-DKL showed the best performance on most of the design spaces.

**Does big model generalize better?** We have witnessed a lot of news reporting large language models showing better zero/few-shot performance, then one interesting question to ask is how transformer model size affects the sample efficiency of our pre-trained model. To answer that question, we increased the model size to the level of BERT (Devlin et al., 2019) with 768-dimensional embedding, 12-heads attention, and 12 transformer layers, the model now has more than **30 million** trainable parameters.

Given only one GPU to use, the large model is very slow to train. It took us about 50 minutes to train for one epoch, so we didn't follow the standard BO procedure as in previous sections. Firstly, GP was not used, we only pre-trained the model with MSE loss for 250 epochs, the model with linear output layer was directly used as the surrogate model; secondly, there's **no fine-tuning** during target task optimization, instead, we only used the pre-trained multi-source FT-Transformer for *zero-shot prediction* on the `meta-test` dataset. We sorted the zero-shot prediction result and the hyperparameter configurations with the top 100 predictions was recommended one by one for the 100 target task iterations. With this *batched zero-shot optimization*, the target task can be done much faster than BO with smaller FT-DKL.

We compared the batched zero-shot optimization of our previously reported pre-trained FT-DKL and FSBO, and we also performed batched zero-shot optimization for the smaller FT-Transformer used for FT-DKL. The result of average rank and regret can be seen in Figure 3.

As can be seen in Figure 3, with equal-sized FT-Transformer, running BO with pre-trained FD-DKL showed better performance than only running batched zero-shot optimization; however, with a much larger multi-source FT-Transformer, the performance of zero-shot optimization would be comparable to the result of BO.

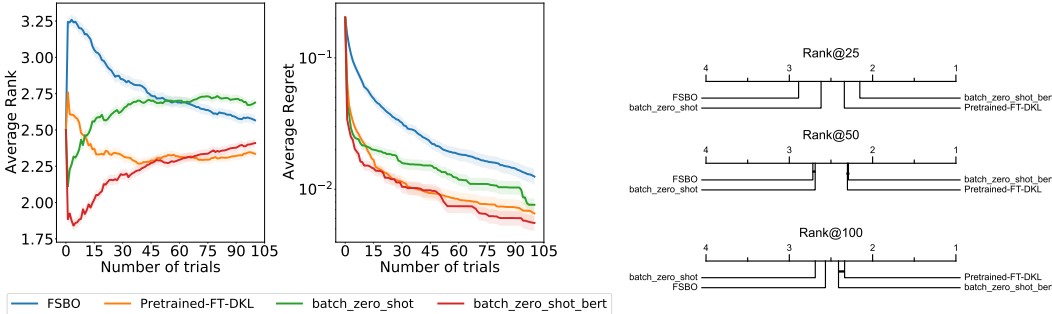

Figure 3: Comparisons of normalized regret and average ranks across all search spaces.

## 5.3 WIRELESS NETWORK OPTIMIZATION

In this section, we evaluate our pre-training strategy in real-world applications. The task is to optimize the parameters of a cellular network. A cellular network consists of many wireless cells and provides the infrastructure for modern mobile communications. Each wireless cell has many parameters (e.g., the antenna angle, the transmission power) that need to be optimized to adapt to its surrounding wireless environment. By optimizing the parameters, the performance (e.g., data throughput) of the network can be improved to provide better user experience.

Due to the heterogeneous wireless environment of a cellular network, different cells have different optimal parameter configurations. Moreover, tuning parameters of one cell affects the performance of its neighbors due to the inter-cell interference. Therefore, parameters of different cells within a network should be jointly optimized.

**Source and Target Task.** In our experiment, the network consisted of 23 cells, where 69 parameters need to be optimized jointly to increase the overall network throughput. In the source task, part of the network (i.e., 16 cells with 48 parameters) had been optimized historically, and we used the data from the source task to pre-train the surrogate model, and transferred to the 69-parameter optimization task to boost the optimization efficiency. The experiments were conducted on an industrial-level simulator that imitated the behavior of a real cellular network.

**Search Space.** The search range of each parameter was an ordered set of integers with step size of 1. Among the 69 parameters, 46 parameters had a search range of size 41, and 23 parameters had a search range of size 26.

**Surrogate Models.** We compared the optimization performance of the following strategy/surrogate models on the 69-parameter optimization task.

- **Random:** A uniformly random sampling was performed within the 69-parameter search space.
- **Random Forest:** A random forest regressor with 100 estimators was used as the surrogate model *without* pre-training, and was initialized by 70 random samples.
- **Gaussian Process:** A Gaussian process model with combined linear and Matern kernel was used *without* pre-training, and initialized by 70 random samples.
- **Pretrained-FT-DKL:** The proposed model was pre-trained with 1800 samples collected from previous 48-parameter BO experiments. Among the 1800 samples, 1000 samples were obtained by uniformly random sampling from the 48-parameter search space, and 800 samples were the BO traces with RF and GP as surrogate models and 400 samples were collected for each model. The optimization on the target task was initialized by 5 random samples.

**Setup.** The exploration budget of each run was 400 iterations, and 5 runs were performed for each surrogate model. The mean and variance over different runs were recorded.

The performance of the surrogate models over different runs are presented in Figure 4, and the numerical values are reported in Table 5. The regret is the negative of the 23-cell-network throughput and the value should be minimized as small as possible. It is clear from the plot that our pre-training strategy boosted the optimization efficiency dramatically compared to the other baselines. After

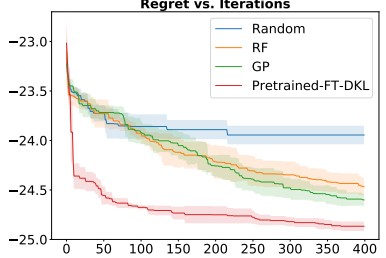

Figure 4: Regret vs. Number of iterations

| Algorithms | Regret@80 | Regret@400 |
|---|---|---|
| Random | -23.853 | -23.945 |
| RF | -23.838 | -24.466 |
| GP | -23.840 | -24.601 |
| **Pretrained-FT-DKL** | **-24.633** | **-24.867** |

Figure 5: Regret by iteration 80 and 400

only a few initial random samples, the Pretrained-FT-DKL model quickly located its search to a region with low regret. The achieved performance of Pretrained-FT-DKL by 80 iterations already surpassed RF and GP's final performance with 400 iterations, which indicated that our strategy boosted the optimization efficiency by around five times.

## 6 LIMITATION AND DISCUSSION

Overall, we believe that our proposed method represents a new paradigm of transfer learning in BO, however, there are still several limitations of our work to be overcome.

The main issue is about the training overhead of Transformers. Even with GPUs, transformer training is still time-consuming compared to GP and traditional deep kernel GP. Although the excellent zero/few-shot performance alleviates the requirement for long-time BO iterations, we still need hardware-friendly Transformers and Transformer-friendly hardware for more widely application of our method.

Secondly, as shown in (Ober et al., 2021), it's even more easier to over-fit a deep kernel model than to over-fit a traditional neural network. The over-fitting can be addressed by introducing Lipschitz constraint in neural works (Liu et al., 2020). However it's still unclear how Lipschitz constraints can be efficiently introduced to Transformers.

Finally, we believe that the quality of the source data is very important. Currently, we use shared embedding vector for common features, and we determine whether or not two source datasets have common features by matching their parameter names, so it's possible that our system can be misguided if a malicious user uploads a dataset with the same parameter names but completely irrelevant or adversarial parameter values.

## 7 CONCLUSION

In this paper, we introduced a simple deep kernel model with multi-source FT-Transformer. The model can be used to jointly pre-train multiple heterogeneous source datasets and can be efficiently fine-tuned for downstream target task with unaligned parameter space. We tested the proposed FT-DKL model on three synthetic and real-world benchmarks and found the model to be highly sample-efficient. We also found that a bigger surrogate model that matched the size of BERT showed even better zero-shot optimization performance. We believe that our research paves the way toward a more unified surrogate model for Bayesian optimization.

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

# A APPENDIX

## A.1 AN ILLUSTRATIVE EXAMPLE OF THE MULTI-SOURCE FT-TRANSFORMER

Assume there are overall five parameters to be optimized: $\mathbf{x} = [x_1, x_2, x_3, x_4, x_5]^\top$, where in the source datasets, only the first four parameters have been used and parameter $x_5$ only occurs in the target optimization task.

Assume we now have two tasks. Task 1 has parameters $\mathbf{x}^1 = [x_1, x_2]^\top$ and task 2 has parameters $\mathbf{x}^2 = [x_2, x_3, x_4]^\top$. Therefore, each example of parameters $\boldsymbol{x}^{(i)} \in \mathbb{R}^2$ in task 1, whereas $\boldsymbol{x}^{(i)} \in \mathbb{R}^3$ in task 2. Examples of response $y^{(i)} \in \mathbb{R}$ are the same in both tasks.

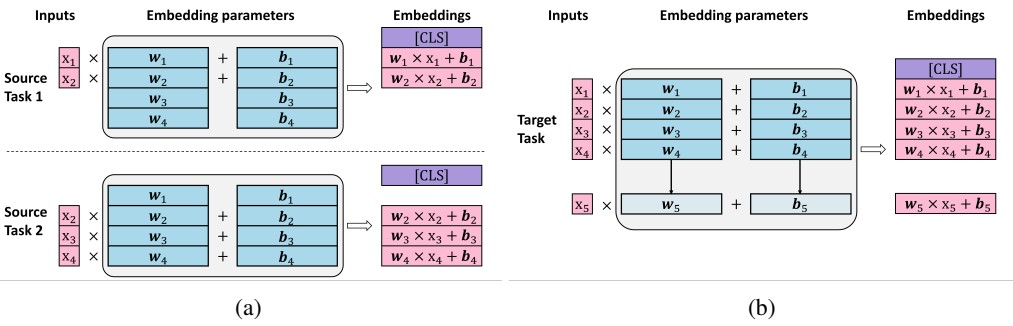

(a)                                (b)

Figure 6: Illustration of feature tokenizer in our joint training framework. (a) In pre-training, each variable in examples is encoded using its corresponding embedding parameters; (b) For a target example, embedding parameters of input variable that exists in the training tasks would be directly adopted. Those of new input variables would be constructed in a mixed-up manner: we first randomly sample two existing embedding parameters $\boldsymbol{w}_i$ and $\boldsymbol{w}_j$, then use their convex combination $\alpha\boldsymbol{w}_i + (1 - \alpha)\boldsymbol{w}_j$, where $\alpha \sim \mathcal{U}(0, 1)$, as their embedding parameters.

During pre-trainining, we construct Feature Tokenizer with three trainable parameters:

- Classification token embedding $\texttt{[CLS]} \in \mathbb{R}^{d_e}$
- Weight embedding $\boldsymbol{W} = [\boldsymbol{w}_1, \boldsymbol{w}_2, \boldsymbol{w}_3, \boldsymbol{w}_4]^\top \in \mathbb{R}^{4 \times d_e}$
- Bias embedding $\boldsymbol{B} = [\boldsymbol{b}_1, \boldsymbol{b}_2, \boldsymbol{b}_3, \boldsymbol{b}_4]^\top \in \mathbb{R}^{4 \times d_e}$

The forward-pass of the Feature tokenizer would be

- Task 1: $\mathrm{FT}(\boldsymbol{x}^{(i)}) = \mathrm{stack}(\texttt{[CLS]}, \boldsymbol{w}_1 \times \boldsymbol{x}_1^{(i)} + \boldsymbol{b}_1, \boldsymbol{w}_2 \times \boldsymbol{x}_2^{(i)} + \boldsymbol{b}_2)$;
- Task 2: $\mathrm{FT}(\boldsymbol{x}^{(i)}) = \mathrm{stack}(\texttt{[CLS]}, \boldsymbol{w}_2 \times \boldsymbol{x}_2^{(i)} + \boldsymbol{b}_2, \boldsymbol{w}_3 \times \boldsymbol{x}_3^{(i)} + \boldsymbol{b}_3, \boldsymbol{w}_4 \times \boldsymbol{x}_4^{(i)} + \boldsymbol{b}_4)$.

The tokenized features of $\boldsymbol{x}^{(i)}$ would be fed into the shared transformer layer and the $\texttt{[CLS]}$ embedding of the output embedding $\boldsymbol{z}^{(i)}$ would be used as the output embedding

- $\boldsymbol{z}^{(i)} = \mathrm{Transformer}(\mathrm{FT}(\boldsymbol{x}^{(i)})) \in \mathbb{R}^{d_e}$

In this way, datasets with heterogeneous design space can be jointly trained, the output embedding can be fed into linear layer for MSE training, and Gaussian process layer for ELBO training.

## A.2 HYPER-PARAMETERS AND TRAINING DETAILS OF THE MULTI-SOURCE FT-DKL MODEL

Unless specified, the following hyper-parameters were used.

- 128 dimension embedding
- three transformer layers
- 512 dimension feed-forward MLP

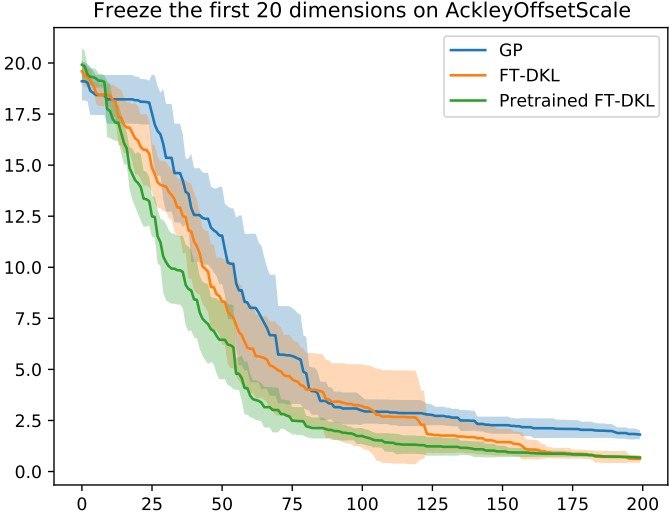

Figure 7: Compare the Pretrained FT-DKL vs Gaussian process on scaled and offsetted 30D-Ackley function, where the first 20 dimensions were fixed to the optimal values. We can see that it is easier to optimize this 10-dimensional function compared to the 30-D modified Ackley function; however, FT-DKL with pre-training still outperformed Gaussian process and un-pretrained FT-DKL.

- learning rate for Transformer and embedding layers: 1e-5
- learning rate for variational parameters with natural gradient: 0.1
- learning rate for Gaussian process kernels and inducing point: 1e-3

We used 128 inducing points for the Ackley and the wireless network optimization problems, and 512 inducing points for HPO-B optimization. We used AdamW optimizer to optimize all parameters except for the variational parameters where natural gradient descent was used.

As the model was firstly pre-trained with linear output layer and MSE loss, we found that commonly used RBF-like kernels generally have worse ELBO loss, so we used an additive kernel with a mix of linear and matern3/2 kernel. The variance hyper-parameter of the linear kernel was initialized as $\text{ceil}(v_l)$, where $v_l$ is the variance of the weight of linear output layer optimized with MSE loss. Also, during the training of Gaussian process with ELBO, we removed the dropout and layer-normalization of FT-Transformer to improve training stability.

We used `StandardScaler` from `sklearn` to normalize input for the experiment of modified Ackley and wireless network optimization, and the `QuantileTransformer` for the HPO-B benchmark. Generally, we found that `QuantileTransformer` used with FT-Transformer lead to better prediction accuracy; however, when used with Gaussian process, `QuantileTransformer` might cause very bad in-between uncertainty prediction. That's not a big issue for the HPO-B benchmark, as there are more than three million source data points for pre-training.

### A.3 ADDITIONAL EXPERIMENTAL RESULT FOR THE ACKLEY FUNCTION OPTIMIZATION

To optimize the 30-dimensional Ackley function with input scaling and offset, we firstly performed Evolutionary optimization on the 20-D AckleyScaleOffset function, and then during optimization of the 30-D function, we fixed the first 20 dimensions to the optimal value of the 20-D function and only optimized the rest 10 dimensions.

We compared Gaussian process and FT-DKL against our pretrained FT-DKL model. As shown in Figure 7, we can see that although GP and FT-DKL was able to converge to near-optimal solution when only optimizing 10 dimensions, it was still outperformed by the pre-trained FT-DKL.

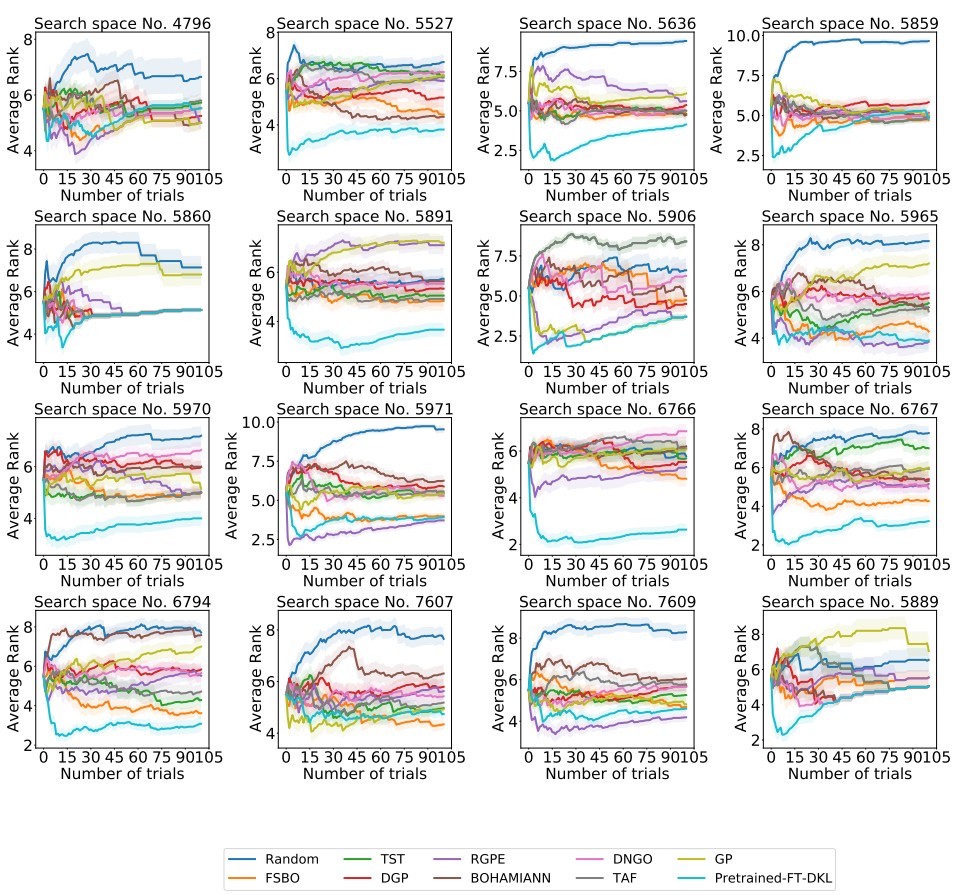

Figure 8: Per-space average rank comparison on HPO-B

## A.4 PER-SPACE RESULT OF HPO-B EXPERIMENTS

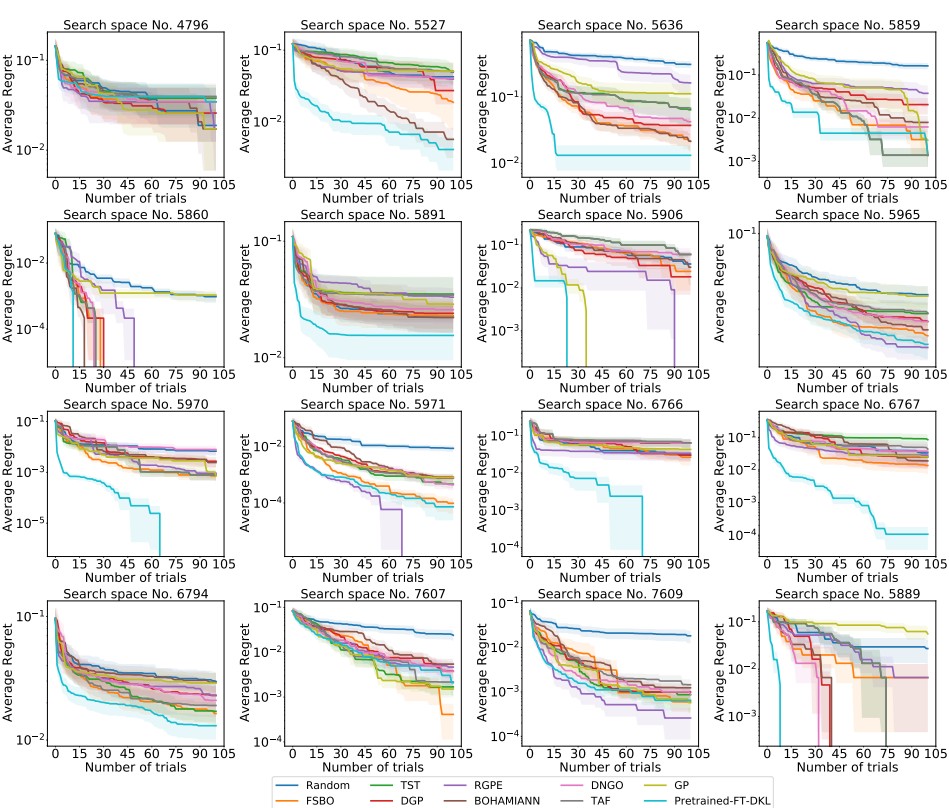

Figure 9: Per-space normalized regret comparison on HPO-B

