# OpenReview forum: "Efficient Bayesian Optimization with Deep Kernel Learning and Transformer Pre-trained on Muliple Heterogeneous Datasets"
_ICLR.cc/2023/Conference — Submitted to ICLR 2023_

### Official Review · Reviewer_cAwj · 2022-10-21

**Confidence:** 5
**Correctness:** 2
**Technical Novelty And Significance:** 2
**Empirical Novelty And Significance:** 2
**Recommendation:** 5

**Clarity, Quality, Novelty And Reproducibility:**

The method is clearly described. The novelty is limited. The community already considered the use of SV-GPs, DKL-GPs and transformer architectures. Therefore, the novelty is the combination of these components into one new method. The main novelty is the claimed learning across search spaces with shared hyperparameters.

My main concern is with the empirical evaluation, especially on HPO-B. First of all, there is no comparison with OPTFormer, another HPO method which is based on transformers. Second, while the method technically trains on all data, we have no empirical evidence that this is useful. It would be great to compare against a version that was trained only on the data from the same search space. Furthermore, HPO-B has many similar search spaces. E.g., there are 6 `ranger` search spaces which only differ in the number of hyperparameters. Since HPO-B is based on OpenML which mainly reports results for UCI datasets, most likely the same datasets have been trained for all `ranger` search spaces. Since no measures have been taken to ensure that a test dataset is not a train dataset on a very similar search space, the hyperparameter task can be very simple for a method that has access to all data. Besides that, it is free additional data that the baselines don't have for unrealistic reasons. In the end, for each of the different search spaces a hyperparameter must have been set (probably the default value). What I would like to see instead is an experiment as used in the text as an example. Use information from RandomForest and LightGBM to boost performance on XGBoost. Right now, it is impossible to understand where the effect is coming from. Third, the overhead was not taken into account. As the authors pointed out, updating their model is very expensive. It is not unlikely that a single update might be more expensive than the function evaluation itself. This could mean that in practice, random search could perform better if both methods get the same wall clock time budget. Therefore, I would like to see all plots where wall clock time is used on the x-axis.

Adding to the authors' discussion section: hyperparameter name matching is extremely difficult and a malicious user won't be the biggest problem (what would be their motivation in the first place?). Just dealing with different names for the same hyperparameter can be challenging (e.g., `lr` vs. `learning_rate` vs. `hp_lr`). If perfect matching is crucial, any blackbox system will mostly fail in practice. It would be great to understand how important that is. Does the method still work if there is no (perfect) matching?

**Strength And Weaknesses:**

**Strengths**
- Interesting idea
- Many experiments

**Weaknesses**
- Experimental setup (details below)

**Summary Of The Paper:**

The authors propose the use of sparse-variational GPs with a deep kernel and a transformer architecture to conduct hyperparameter optimization across different search spaces. They conduct an empirical comparison on three different tasks against most of the state-of-the-art.

**Summary Of The Review:**

This work presents an interesting work that attempts to leverage knowledge from observations made on similar search spaces. There are several concerns regarding the experimental protocol: missing related work, possible test leakage, lack of clarity of the impact of using the additional data and what is causing it, and whether the improvements hold when considering wall clock time.

---

### Official Review · Reviewer_DKdC · 2022-10-24

**Confidence:** 4
**Correctness:** 3
**Technical Novelty And Significance:** 2
**Empirical Novelty And Significance:** 2
**Recommendation:** 3

**Clarity, Quality, Novelty And Reproducibility:**

I have a question on the following sentence:

> The tokenization process for categorical data is implemented by a look-up table.

If we use a look-up table (I know you did not consider a categorical variable in this work), how can you handle a heterogeneous dataset? I think this sentence makes a reader confusing, because a look-up table has a fixed set of key and values. It implies that you cannot deal with a heterogenous dataset.

I also have a question on the mix-up initialization. Since embedding vectors are not bounded, this initialization leads us to initialize a set of parameters wrongly. How do you think of this issue?

Furthermore, I think the following sentence is technically unimportant:

> Finally, we believe that the quality of the source data is very important. Currently, we use shared embedding vector for common features, and we determine whether or not two source datasets have common features by matching their parameter names, so it’s possible that our system can be misguided if a malicious user uploads a dataset with the same parameter names but completely irrelevant or adversarial parameter values.

How do you think? Is it really a serious problem? Along with this question, Section 6 should be revised. The limitations and discussion described in this section are not meaningful. I am wondering if there are more important and more significant limitations and thinks to discuss.

**Strength And Weaknesses:**

From now, I will describe the strengths of this work first.

## Strengths

I think the most important strength of this work is that the proposed method is capable of handling diverse datasets defined on different search spaces.

Also, a simple method based on a mix-up initialization technique is quite impressive. I think that the simplicity of the algorithm is quite important in terms of reproducibility and versatility.

However, it has some limitations regarding presentation and reasoning. Moreover, the novelty of this work is also my concern.

## Weaknesses

First of all, I think the presentation of this work can be improved by revising sentences, elaborating explanations, and removing redundant contents. Although an introduction section is clear, a background section needs to be improved. For example, the explanation of Gaussian process regression is quite redundant. I think that the writing about deep kernel learning for Gaussian process regression should be described. I am curious how deep kernel learning is used in this work and which part of deep kernel learning is significant in proposing the algorithm. The description on Bayesian optimization is similar. Since I am quite familiar with Gaussian processes and Bayesian optimization, I can understand Section 3 even though the background section is not kind, but I think most readers who are not familiar with these topics are difficult to understand this work. Also, the description of feature-tokenizer Transformer is not enough. I am curious about the architecture of the Transformer and its details, e.g., layers, layer normalization, and inputs/outputs. Although the feature-tokenizer Transformer is directly adopted in your work, it should be carefully discussed. Also, Algorithm 1 seems like the plain sentence rather than a pseudo code. I think that Figure 6 is more helpful to understand this work. This figure should be located on the main manuscript.

More importantly, reasoning on why your algorithm works well is not sufficient. I carefully read Section 3.2, but it does not explain why it works well. As describe above, the mix-up initialization strategy is very clear and it seems effective. However, the reasoning is missing. I would like to hear why it is effective.

I would not like to mention the novelty of this work directly. However, the proposed method is a combination of existing work. I emphasize that this fact is not important, but you need to describe why you choose such a combination and which part of the algorithm shows a certain circumstance.

**Summary Of The Paper:**

This work suggests a Bayesian optimization strategy with deep kernel learning and a feature-tokenizer Transformer, where we are given multiple heterogeneous datasets. By utilizing deep kernel learning and the feature-tokenizer Transformer, it initializes a set of parameters for the feature-tokenizer Transformer in order to adopt to a target task. Finally, the authors demonstrate the experimental results to show the effectiveness of the proposed method.

**Summary Of The Review:**

Therefore, the current version is not enough to be accepted in ICLR 2023; please see the aforementioned comments.

---

### Official Review · Reviewer_xRh6 · 2022-10-25

**Confidence:** 3
**Correctness:** 3
**Technical Novelty And Significance:** 2
**Empirical Novelty And Significance:** 3
**Recommendation:** 6

**Clarity, Quality, Novelty And Reproducibility:**

The paper is well written. Most of the individual components (like the FT-transformer) are already existing in the literature and employed directly for Bayesian optimization framework.

**Strength And Weaknesses:**

- The paper considers an important problem since multiple related optimization tasks are being solved quite commonly nowadays (for e.g. hyper-parameter optimization of ML models).

- The proposed approach performs well on the benchmarks employed for experimentation and analysis.

- The illustrative example in the appendix is really nice and useful to understand the transformer architecture employed in the paper.


However, I have few questions and suggestions to understand some of the details better and that will hopefully improve the paper's contributions:


- Although there is a passing remark about generalization issues with deep kernel learning (especially in the small data setting) in limitations section, this should be discussed in the beginning with more elaborate discussion. One way to test the surrogate model independently of the BO loop is to construct a standard train-test scenario and evaluate the model log-likelihood or mean squared error on the test set.

- It is mentioned that most of the existing meta-learning/transfer learning approaches in Bayesian optimization are not applicable for varying input spaces. However, one straight-forward way to employ them is to pick a large enough search space that contains a union of input spaces from all the source tasks. What would be the advantages of the proposed approach over this simple baseline?

- The mix-up initialization for the unseen task parameters is defined without proper justification. Please add more details about this choice including some principles that make it pertinent to the problem setting.

- It is surprising that only one benchmark in the experimental evaluation has more than one source tasks (i.e. only HPO-B has 758 source tasks while other has only 1 source task). This seems to limit the analysis of the proposed approach which would be presumably most effective with multiple sources. It is absolutely not necessary to do this in the rebuttal period but if possible, please consider adding another benchmark with multiple source tasks to show the effectiveness of the proposed approach in a more robust way.


**Summary Of The Paper:**

The paper considers the problem of black-box optimization of expensive black-box functions when data from multiple related optimization tasks is available. The key contribution is to employ a Feature-Tokenizer (FT) transformer to learn from multiple source datasets where each dataset might be defined over different input space. A deep kernel learning based gaussian process surrogate model is trained on top of the transformer representation to be plugged in with standard Bayesian optimization techniques on the target task. Experiments are performed on 3 benchmarks including hyper-parameter optimization and wireless network optimization.


**Summary Of The Review:**

Overall, I think the paper considers an important problem and provides a simple solution based on existing ideas. However, some of the design choices and the experimental analysis require more elaborate discussion to improve the quality of the work.

---

### Decision · Program_Chairs · 2023-01-20

**Decision:**

Reject

**Justification For Why Not Higher Score:**

- Lack of novelty
- Issues in experimental evaluation

**Justification For Why Not Lower Score:**

N/A

**Metareview: Summary, Strengths And Weaknesses:**

The paper considers the problem of solving several related optimization problems with different search spaces. It uses a feature-tokenizer transformer to embed these optimization tasks into a single vector space, over which it uses sparse variational Gaussian process regression to perform inference. It then uses this Gaussian process predictive method within a Bayesian optimization framework to guide experiments. It finds significant improvement over baseline methods in hyperparameter optimization and wireless network optimization.

Strengths
- The paper considers an important problem --- it is common to need to solve several related optimization tasks with diverse datasets defined over different search spaces
- Method performs well empirically against baselines across many experiments

Weaknesses
- Lack of novelty --- the proposed method is a straightforward combination of Feature-Tokenizer transformers, sparse variational Gaussian processes, and Bayesian optimization.

- Lack of comparison against a method that trains only on data from the same search space. Since the claimed value of this method is in the ability to train on different search spaces, we should see that it outperforms methods that don't have this capability. Moreover, the evaluations would seem to allow a method that can use data only from the same search space to do quite well. It isn't clear whether the proposed method works better than baselines because it simply has data from the same search space and different tasks, or it is truly getting benefit from looking at different search spaces. This issue is articulated well by reviewer cAwj.

- Lack of discussion of computational overhead.

- Lack of comparison against OPTFormer (recent HPO method based on transformers, Chen et al. 2022)

- The quality of the writing is low

Notably, the authors did not provide a rebuttal to the reviews.